# Reinventing Therapeutic Proteins: Mining a Treasure of New Therapies

**Sarfaraz K. Niazi** [1,*] and **Zamara Mariam** [2]

1   College of Pharmacy, University of Illinois, Chicago, IL 60612, USA
2   School of Interdisciplinary Engineering & Sciences, National University of Sciences & Technology, Islamabad 476415, Pakistan
*   Correspondence: sniazi3@uic.edu; Tel.: +1-312-297-0000

**Abstract:** Reinventing approved therapeutic proteins for a new dose, a new formulation, a new route of administration, an improved safety profile, a new indication, or a new conjugate with a drug or a radioactive source is a creative approach to benefit from the billions spent on developing new therapeutic proteins. These new opportunities were created only recently with the arrival of AI/ML tools and high throughput screening technologies. Furthermore, the complex nature of proteins offers mining opportunities that are not possible with chemical drugs; bringing in newer therapies without spending billions makes this path highly lucrative financially while serving the dire needs of humanity. This paper analyzes several practical reinventing approaches and suggests regulatory strategies to reduce development costs significantly. This should enable the entry of hundreds of new therapies at affordable costs.

**Keywords:** therapeutic proteins; recombinant proteins; repurposing; reinventing; drug–antibody combinations; AI/ML; efficacy improvement; immunogenicity; artificial intelligence; mRNA; high-throughput analysis

## 1. Introduction

The famous quote of the 1988 Nobel Laureate in Medicine, James Black [1], that 'the best way to discover a new drug is to start with an old one,' sets the theme of reinventing therapeutic proteins to capitalize on their multibillion-dollar cost. Fourteen years from their development [2] is a novel approach to introduce biological therapies based on approved therapeutic proteins' safety and efficacy claims. It could be a new dose, a new delivery system, a new route of administration, a new indication, or a new combination with other therapeutic proteins, chemical drugs, or radiation sources.

These reinventing options are widely adopted [3]—mainly when treating rare and neglected diseases with limited patients. However, reinventing also helps in situations where faster development is critical, as happened during the COVID-19 or Ebola outbreaks which led to a vigorous push to repurpose the use of multiple antibodies, as there was no time to wait for a new drug.

The main advantage of the reinventing strategy is that its safety and manufacturing processes are already established, which reduces the need for extensive research and development, including preclinical testing, thus, taking the reinvented entity direct to phase III testing in most cases [4]. This is a major cost saving, allowing a continued amortization of the initial development cost.

Drug reinventing often arrives serendipitously from the surprising effects observed for an approved drug. As 'chance favors only the prepared mind [5], serendipity has produced significant advances in the history of medicine and selective optimization of side activities of drug molecules for generating new drugs [6]. Examples of chemical drugs have been repurposed for benign prostatic hyperplasia, angina, sedation, nausea, and

insomnia; later, they were repurposed for use in hair loss, erectile dysfunction, and leprosy, respectively [7]. Examples of serendipitous discovery include sildenafil, intended for the treatment of hypertension and ended up as the most popular male erectile dysfunction treatment; dimethyl fumarate, developed to treat multiple sclerosis [8], ended up treating psoriasis [9]; or the antiviral drug remdesivir under testing to treat Ebola infection, ended up treating COVID-19 [10].

Beyond serendipity, we can reinvent new drugs using technologies such as drug–target interactions (DTI). AI-driven in silico tools significantly helps DTI mapping for drug reinvention. This technique has played a vital role in identifying potential therapeutics during the COVID-19 pandemic. A deep learning model trained on drug–target interaction (DTI), molecule transformer–drug–target interaction (MT-DTI), has uncovered alternate uses of available drugs: atazanavir and remdesivir efavirenz, ritonavir, and dolutegravir as inhibitors against SARS-CoV-2 protein [11]. CATNIP, a machine learning (ML) model for drug repurposing, uses similarity data of the molecules based on their structure, target, and pathway information for drug reinvention [12]. Besides identifying clinical targets, AI-based models can also identify adverse effects of therapeutics. For instance, chemical fingerprint data were used to develop a model which predicted that 22 FDA-approved drugs have potential contributions to heart failure. Later, experimental validation confirmed that 8 out of 22 anticipated therapeutics had cardioprotective activities [13].

A newer [14] approach for drug repurposing involves two-stage prediction and machine learning. First, diseases are clustered by gene expression because similar altered gene expression patterns imply critical pathways shared in different disease conditions. Next, drug efficacy is assessed by the reversibility of abnormal gene expression, and results are clustered to identify repurposing targets. Finally, the functions of affected genes are analyzed to examine consistency with expected drug efficacy.

Adding a new indication is one of the fastest routes to reinventing therapeutic proteins because of the diversity of pharmacologic responses of therapeutic proteins; they need to be discovered. It is anticipated that new indications can be added to most approved therapeutic proteins, opening a vast treasure of therapies at a much-reduced development cost since the therapeutic protein's safety is already established. Examples of therapeutic proteins that have received new indications recently include Actemra (tocilizumab), Adcetris (brentuximab vedotin), Dupixent (dupilumab), Enhertu (fam-trastuzumab deruxtecan-nxki), Eylea (aflibercept), Hadlima (adalimumab-bwwd), Imfinzi (durvalumab), Jemperli (dostarlimab-gxly), Kevzara (sarilumab), Keytruda (pembrolizumab), Libtayo (cemiplimab-rwlc), Takhzyro (lanadelumab-flyo), Tecentriq (atezolizumab), Tezspire (tezepelumab-ekko), Trodelvy (sacituzumab govitecan-hziy), and Trogarzo (ibalizumab-uiyk) [15]. A biosimilar can also obtain a new indication if not protected by a patent, which significantly expands the drug's utility.

## 2. Understanding Therapeutic Proteins

"Therapeutic protein" refers to recombinant DNA (rDNA) products that join DNA from different species and subsequently insert the hybrid DNA into a host cell, often a bacterium or mammalian cell, to express the target protein. UC San Francisco and Stanford researchers created this molecular chimera in 1972 [16]. Stanley Cohen of Stanford and Herbert Boyer of UCSF received the US patent in 1980. On 26 July 1974, ten researchers, including six future Nobel Laureates (James Watson, Paul Berg, Stanley Cohen, David Baltimore, Ronald Davis, and Daniel Nathans), wrote a letter in Science [17] urging that the NIH regulate recombinant DNA technology.

The first rDNA product came in 1982 when the rDNA insulin was approved [18]; now, hundreds of recombinant proteins are approved by regulatory agencies [19]. Examples of this diverse class of compounds include interferons, cytokines, interleukins, thrombocytes, growth factors, coagulation factors, blood factors, anticoagulants, Fc fusion proteins, monoclonal antibodies, etc. [20]. The global biologics market size is expected to reach around USD 719.94 billion by 2030, valued at USD 366.50 billion in 2021 and growing at a

CAGR of 7.15% from 2022 to 2030. The current market of therapeutic proteins exceeds USD 380 billion [21].

Therapeutic proteins replace a protein that is abnormal or deficient in a particular disease or augments the body's supply of a beneficial protein to help reduce the impact of disease or chemotherapy. Genetically engineered proteins can closely resemble the natural proteins they replace or be enhanced by adding sugars or other molecules that extend the protein's duration of activity.

For regulatory approval, the FDA treats alpha amino acid polymer with 40 or fewer amino acids as a peptide, not a protein [22]. It is regulated as a drug under the FD&C Act rather than the Public Health Service (PHS) Act which controls biological drugs. Other definitions of peptide define the range of amino acids from 2 to 50 [23].

The unique properties of proteins arrive from the long chain of amino acids in therapeutic proteins that fold into a three-dimensional (3D) structure of domains that attach to receptors, resulting in pharmacological responses that can be extended to the toxicological response. In addition, proteins are, by nature, immunogenic, a property that can also be modulated by altering the structure.

Polypeptide chains are combinations of 20 different types of amino acids resulting in the production of numerous proteins due to the high degree of freedom, as pointed out by Cyrus Levinthal in 1969. Suppose we account for only three states of each bond for an amino acid sequence with 101 residues, 100 peptide bonds, and 199 distinct phi and psi bond angles. In that case, a protein can fold into a maximum of $3^{100} = 5 \times 10^{47}$ possible conformations. It will take approximately $10^{27}$ years to test all the possibilities at a protein sampling rate of $3 \times 10^{20}$ per year [24,25]. This paradox of the natural folding of proteins was only recently resolved, claiming that as proteins fold into native states, they mostly reach a state of minimum energy and maximum stability. This observation will lead to the use of AI-based protein structure prediction and its confidence in repeatability. This will become a critical exercise in evaluating the safety of copies of proteins as biosimilars, as discussed below.

The high flexibility, structural plasticity, and specificity of intrinsically dynamic systems determine receptor binding modes, pharmacokinetics (PK), pharmacodynamics (PD), bioavailability, drug target, and anti-target protein interactions, and their relative affinity [26]. Briefly, the possible structural diversity of domains suggests that a protein molecule could have multiple modes of action and, thus, therapeutic applications (Figure 1).

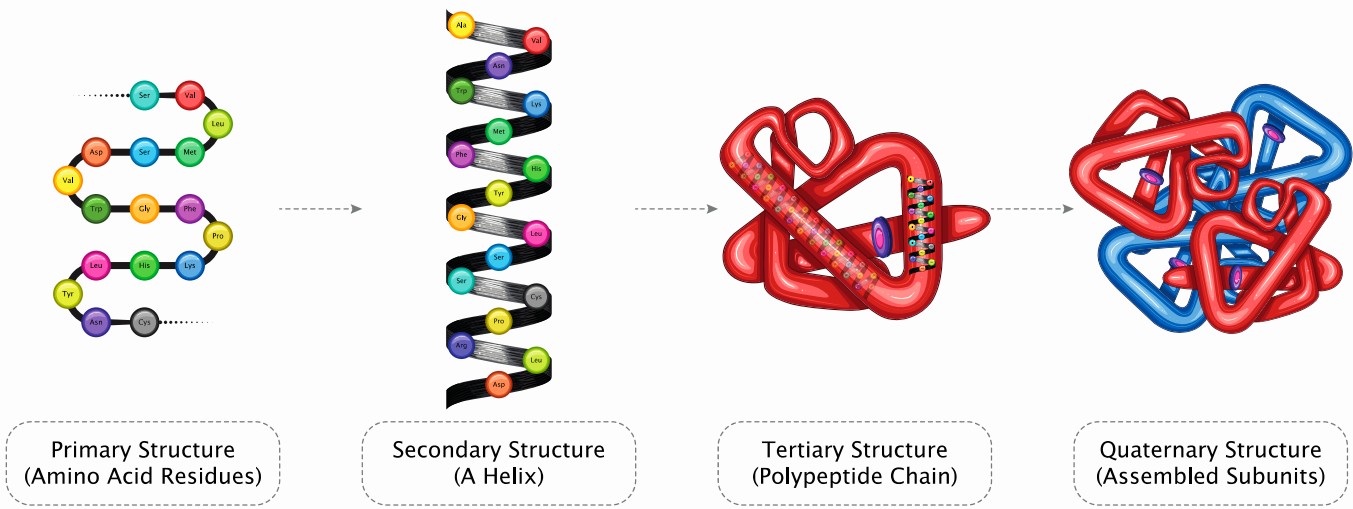

**Figure 1.** Amino acid chains form a secondary structure (helix), resulting in a polypeptide chain folding to form proteins (Licensed from Shutterstock).

The 3D structure of proteins defines their functions; specific domains interact with receptors, triggering pharmacological and toxicological responses. The biological assay

reflects the known mechanism and thus serves as a link to clinical activity. Therefore, using relevant biological assay(s) of appropriate precision, accuracy, and sensitivity is essential to confirming no significant functional difference.

A key element of protein structure is the domain resulting from its stable structure that can fold and undergo folding without reference to the rest of the amino acid chain. Domains are not necessarily unique; the same gene can be found in many molecules. The binding domain binds to a specific atom or molecule, such as calcium or DNA. Proteins may have a conformational change as a result of binding. Many proteins depend on their binding domains to work correctly. They are necessary because they aid in the splicing, assembling, and translating proteins [27] (Figure 2).

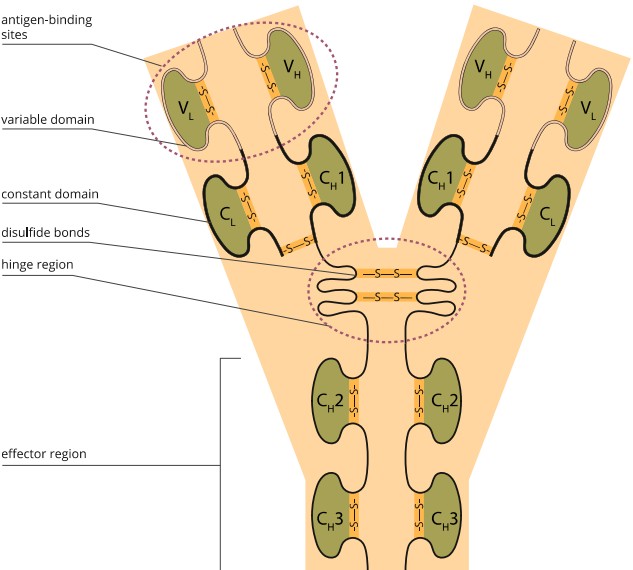

**Figure 2.** Variable and constant antibody domains (Licensed from Shutterstock).

Given the many possible domains, the approved indications of a protein drug only represent a limited activity of the tested or known domains, allowing the discovery of numerous other efficacy profiles. For example, many antibodies were proposed with new indications to control COVID-19 infection [28], and bevacizumab continues to add drug combinations and newer indications in treating age-related macular degeneration [29,30].

Monoclonal antibodies (mAbs) are immunoglobulins that bind to specific protein epitope targets, cancer, and stromal cells, giving them therapeutic properties. The mAb properties of importance are (i) binding affinity to the target antigen; (ii) binding to Fc receptors such as FcγRI, Ia, IIa, IIb, IIIa, IIIb, and FcγRN; (iii) assessment of effector functions such as ADCC and (iv) CDC; (v) molecule characteristics such as charge, pI, hydrophobicity, and glycosylation; and (vi) off-target binding using in silico or in vitro methods such as baculovirus ELISA tools [31,32]. More specifically, for TNFα blockers: C1q, CDC, induction of regulatory macrophage, inhibition of T-cell proliferation (MLR), LTα, MLR, mTNFα, off-target cytokines, reverse signaling, sTNFα, suppression of cytokine secretion, tmTNF-α.

## 3. Reinvention Scope

Advancement in recombinant technology has enabled developers to fine-tune and increase the therapeutic potential of proteins by targeting their structure and function to enhance their disposition half-life, product yield, and purity [33]. Modifying disposition kinetics is also an excellent opportunity to reduce the dosing frequency.

The current reinventing approaches are less serendipitous and more based on rational and systematic approaches; libraries of approved compounds are available from many commercial sources. In addition, several computational and high-content screening methods are currently used to discover new indications for existing molecules [34]. When a hit emerges from a drug reinventing strategy, it can be taken directly into the last phases of clinical trials [35]. However, the side effects of therapeutic proteins can be disease-dependent, unlike chemical drugs, requiring creative approaches to establish safety [36].

## 4. Intellectual Property

A major hindrance in reinventing therapeutic proteins is their intellectual property protection for the gene that expresses the molecule. If a new indication is patented, it will be allowed once the gene patent expires. However, this bar is coming down fast as many therapeutic proteins are now off the patent [37]. In addition, the intellectual property hurdles go beyond gene patents and include process-related patents that can extend the research work to remove any infringement [38].

## 5. Artificial Intelligence (AI) and Machine Learning (ML)

ML uses algorithms that can recognize patterns within a data set that has been further classified. A subfield of ML is deep learning (DL), which engages artificial neural networks (ANNs). These comprise a set of "perceptrons", interconnected sophisticated computing elements mimicking biological neurons with their electrical impulses in the brain [39]. ANNs constitute a set of nodes, each receiving a separate input, ultimately converting them to single or multi-linked outputs using algorithms to solve problems. ANNs involve various types, including multilayer perceptron (MLP) networks, recurrent neural networks (RNNs), and convolutional neural networks (CNNs), which utilize either supervised or unsupervised training procedures [40]. The MLP network provides pattern recognition, optimization aids, process identification, and controls based on training in a single direction to enable universal pattern classifications. RNNs are networks with a closed loop, capable of memorizing and storing information, such as Boltzmann constants and Hopfield networks [41]. CNNs are a series of dynamic systems with local connections characterized by their topology. They have been used in image and video processing, biological system modeling, processing complex brain functions, pattern recognition, and sophisticated signal processing. The more complex forms include Kohonen, RBF, LVQ, counter-propagation, and ADALINE networks.

AI modeling can significantly reduce preclinical work. The prediction of the toxicity of any drug molecule is vital to avoid toxic effects, as predicted by LimTox, pkCSM, admetSAR, and Toxtree, which are available to help reduce the cost [42]. Advanced AI-based approaches look for compounds' similarities or project the compound's toxicity based on input features. The Tox21 Data Challenge organized by the National Institutes of Health, Environmental Protection Agency (EPA), and US Food and Drug Administration (FDA) was an initiative to evaluate several computational techniques to forecast the toxicity of thousands of environmental compounds and drugs; an ML algorithm named DeepTox outperformed all methods by identifying static and dynamic features within the chemical descriptors of the molecules, such as molecular weight (MW) and Van der Waals volume. It could efficiently predict the toxicity of a molecule based on predefined 2500 toxicophore features [43].

Drug–protein interactions can also predict the chances of poly-pharmacology, which is the tendency of a drug molecule to interact with multiple receptors producing off-target adverse effects [44].

Traditional drug discovery projects relying on in vitro high-throughput screening (HTS) involve large investments and sophisticated experimental set-ups, affordable only to big biopharmaceutical companies. In this scenario, the application of efficient state-of-the-art computational methods and modern artificial intelligence (AI)-based algorithms for rapid screening of repurposable chemical space, approved drugs, and natural products

(NPs) with proven pharmacokinetic profiles to identify the initial leads is a powerful option to save resources and time. Structure-based drug repurposing is popular in silico repurposing approach [45].

Developing novel inhibitors against discoidin domain receptor-1 (DDR1) within 46 days and cyclin-dependent kinase-20 (CDK20) as a potential anti-lung-cancer drug within 30 days through AI-driven models is remarkable proof of 'intelligent' drug discovery [46,47]. AI-driven repurposing is many folds faster than the traditional method. The lock and key analogy (Figure 3) demonstrates the main challenges for artificial intelligence (AI) in drug reinvention [48]. In contrast to the conventional method of target search, AI-driven methods enable screening of a larger number of locks (targets) and enable testing of available keys (small molecules) through virtual screening in a shorter period. It enables discovery, development, optimization, reinventing, and in silico testing of the exact key for target molecules.

While a protein may have been identified for a specific pharmacological response, much of the landscape of protein structure as domains remain unexplored; discovering a new key will expand the utility of an approved therapeutic protein (Figure 3).

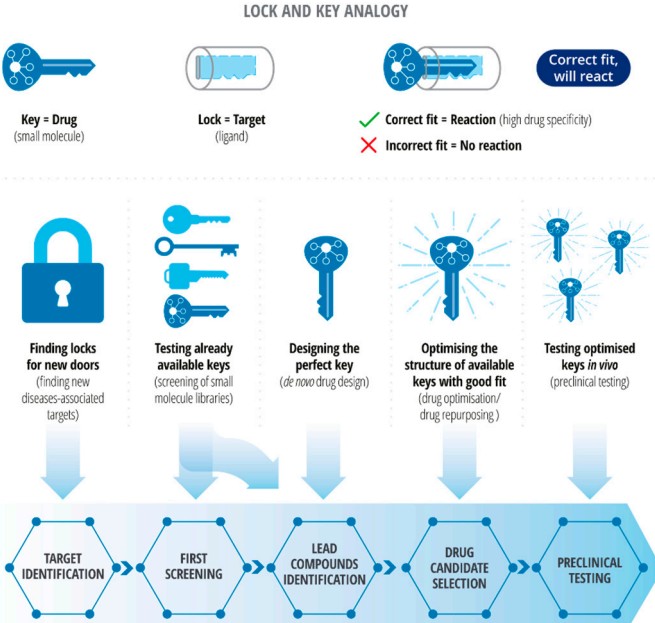

**Figure 3.** Lock and key analogy of challenges for AI in drug discovery [48].

AI-directed methods can further automate lead optimization, improve drug safety, design molecules with specific properties, and scrutinize structural databases to design poly-pharmacological and multi-target agents.

*5.1. Structure Prediction*

Finding domains that can bind starts with a detailed structural analysis. Experimental methods for protein structure identification include X-ray crystallography, nuclear magnetic resonance, cryo-electron microscopy, circular dichroism spectroscopy, etc. [49,50]. However, the testing variability of these methods depends on the quality of samples and precision of equipment, and the results can be compared with the data reported in UniProt and RCSB Protein Databank (PDB); currently, the PDB has approximately 174,825 experimentally derived structures available for comparison [51].

With major advancements in machine learning and AI, template-free protein structure prediction methods have also increased accuracy and reliability of structure prediction methods. Template-free AI models are trained on the sequence and structural data from

openly available databases, i.e., UniProt, RCSB PDB, Uniclust [52], BFD [53], MGnify [54], etc. Highly accurate protein structure prediction tools independent of templates include AlphaFold2 [55], trTosetta [56,57], Robetta [58], RoseTTA Fold [59], ESMFold [60], and OmegaFold [61]. Each algorithm uses a different AI model to predict protein structures from amino acid sequences. For example, AlphaFold2 uses a deep neural network-based approach with over 200 million protein structures openly available in the AlphaFold2 database [62]. trRosetta uses transfer learning with pre-trained deep neural networks; the Robetta server combines ab initio and homology-based methods with machine learning algorithms.

In contrast, RoseTTA Fold combines the strengths of Rosetta and deep neural networks. In addition, ESMFold uses energy-based statistical mechanical and language models, and OmegaFold integrates a protein language model with an end-to-end deep learning framework. These variations allow an orthogonal approach to predict the structure, providing greater reliability of the results. As a result, AI-based structure prediction tools have accelerated the process of therapeutic protein reinvention.

For some proteins, the structure can be predicted through template-based homology modeling, protein threading, and ab initio methods with the assistance of computational tools, i.e., I-TESSER [63,64], SWISS-MODEL [65], MODELLER [66], etc. Despite the significant differences in the specific procedures used by these prediction methods, the underlying steps are similar, including template selection, structure reconstruction, refinement, and analysis [67].

AI-driven retrosynthetic routes [68], phenotypic data or disease data, and molecule network-based algorithms without much structural data are used to design structures that can bind to the interface of targets while controlling their solubility [69,70] and benchmarking antibody discovery through AlphaFold2-enabled molecular docking and simulations [71]. One of the most remarkable events of AI-driven drug discovery was the application of AlphaFold2, PandaOmics [72] in discovering a small molecule target against cyclin-dependent kinase 20 (CK20) with a binding affinity of $9.2 \pm 0.5$ μM ($n = 3$), designed and tested in only 30 days [47].

*5.2. Target Identification*

In the on-target strategy, a new indication of the drug acting through the originally known target is explored since the mechanism of action is expected to retain the same therapeutic effects. In the off-target strategy, new drug uses are identified acting through an unanticipated target; in this case, the mechanism of action is not apparent. Docking and fingerprinting are standard methods.

The use of AI tools in drug–target identification has dramatically improved the efficiency of drug reinventions by enabling the concurrent screening of active compounds and predicting potential drug targets with greater accuracy. AI-based tools have revolutionized how pharmaceutical companies approach discoveries, significantly reducing the time, cost, error, and bias in finding new disease treatments.

High-throughput screening has long been a popular method in drug–target identification. Based on hit and trial, chemical compounds are screened against potential targets to identify compounds with desirable pharmacological properties. More precise target-based screening methods comprise identifying and developing molecules against specific targets, followed by phenotypic screening by screening compounds against cells or tissues. These discovery techniques have previously overcome the needle-in-the-haystack probabilities of such searches.

Under development are many new AI tools for screening active compounds in the search for hit compounds and enhancing the efficiency of the development process [73].

- AtomNet is a convolutional neural network-based tool that applies the concepts of feature locality and hierarchical composition extracted through protein sequence, structure, and function to model bioactivity and chemical interactions of potential drug targets [74]. AtomNet's parent AtomWise has recently enabled the rapid discovery

of drugs against 27 disease targets. DeepDTA is also a deep-learning-based model that uses only sequence information of targets and drugs to predict drug–target interaction binding affinities and potential small molecules as drug candidates from given biological data [75].

- A commercially available natural compounds database and search engine that operates using machine learning, MolPort, when used with quantitative-structure-activity relationship (qsar), analyze the chemical structure and predicts the biological activity of potential targets in the early stages of drug discovery [76].
- Pathway analysis also enables the identification of potential targets. Some crucial biological pathways are available on the Kegg Pathway database [77], which provides insight into a disease mechanism. TargetNet [78] uses this pathways data and protein interaction profiles to predict potential drug targets against a specific disease.
- DeepDock is the most recent AI-driven virtual screening platform with a vast library of small molecules. For example, DeepDock virtual screen results were used to identify 15% active molecules that led to the discovery of novel compounds against the Mpro protease of SARS-CoV2 [79].

*5.3. Molecular Docking*

Identifying structure, functional regions, interaction profiles, and immune system responses are crucial for the success of a therapeutic protein reaching the patients. Therefore, researchers have redirected their attention from conventional drug discovery methods to computational techniques to find new and effective therapeutic agents quickly.

Proteins interact with their receptors to initiate therapeutic effects and manipulate disease mechanisms. For years, fluorescence-based assays, isothermal titration calorimetry (ITC), surface plasmon resonance (SPR), NMR, and other methods have been used to study the binding patterns and thermodynamic effects of drug–target interactions. While highly relevant for characterizing interactions, they are time-consuming, expensive, and resource intensive. Using computational tools in molecular docking has expedited the drug discovery process exponentially, enabling repeated testing with the complexities of the classical method.

Structure-based drug discovery (SBDD) and ligand-based drug discovery (LBDD) both involve the identification of non-covalent interactions using molecular docking in the prediction of novel properties of therapeutic compounds following the lock-and-key hypothesis and induced-fit model [80,81]. In addition, both rely on molecular docking to predict the binding affinity and specificity of small molecules and their targets.

Advancements in computational techniques have led to more precise identification and optimization of binding mode, binding affinity, binding pocket, and solvation effects on drug-target interactions.

- Computer-based tools such as AutoDock Vina [82,83], LigandFit [84], UCSF DOCK [85], and GOLD [86] are widely used.
- Higher binding affinity scores from an in-silico docking analysis of monoclonal antibodies (mAbs) against Alpha and Delta strains of SARS-CoV spike protein suggested that tixagevimab, regdanvimab, and cilgavimab can neutralize most Alpha strains efficiently and bamlanivimab, tixagevimab, and sotrovimab can be effective in suppressing the Delta strain [87]. Venetoclax [88], for treating chronic lymphocytic leukemia, was designed to target the overexpressed BCL-2 protein in cancer cells by binding to its hydrophobic groove. Its development involved optimizing the binding interactions between the drug and BCL-2 through in silico docking studies, highlighting the importance of docking in drug design.

Computational techniques have enabled the targeted discovery of drugs through detailed interaction studies instead of blind docking. However, the precision of these interactions is highly dependent on the pose generation algorithms, binding pocket identification, and scoring functions. In addition, therapeutic targets, such as proteins and ligands, have a large conformational degree of freedom, resulting in extensive data to analyze.

Docking programs sample works through variable methods by treating ligands as flexible or proteins as flexible or/and, in some cases, both as flexible.

- GOLD uses a genetic algorithm, and Autodock Vina uses a grid-based energy approach with a genetic algorithm.
- ICM [89] uses multiple stochastic runs.
- GLIDE SP [90] uses several sampling and scoring methods.
- DeepBSP, an ML-based sampling and evaluation tool, is very useful in generating and ranking profiles close to their respective native structures as a machine learning model-based pose sampling and evaluation [91].
- Identification of the correct view is crucial for higher binding affinity and lower steric hindrance, which can be efficiently achieved through precise AI-based tools. Structure prediction tools such as AlphaFold2 and trRosetta can be integrated with other ML-based approaches to identify and optimize potential poses. One such instance is identifying transition states between the active and inactive conformations of G-protein coupled receptors using multiple ML approaches [92].
- The effectiveness of interaction between the dynamic views and their binding partners can be weighted through scoring systems. Scoring functions are categorized into force-field-based, knowledge-based, and empirical scoring functions.
- Force-field-based scoring functions utilize molecular mechanics to evaluate complex energetic affinities based on their interactions, i.e., weak Van der Waals, electrostatic forces, bond stretching, bending, and torsional angles [93].
- Knowledge-based scoring functions include statistical analysis of distance-dependent atom-pair potentials of protein–ligand or protein–protein complexes generated directly from experimental structures [94,95]. Empirical scoring functions, e.g., LUDI [96], ID-Score [97], and GlideScore [90], are based on empirical data. They correlate binding free energies to weak Van der Waals energy, electrostatic energy, desolvation, entropy, enthalpy, H-bonding, rotational and translational degrees of freedom, polar and lipophilic effects, and hydrophobicity in the form of simple equations to reproduce experimental affinity data.
- These scores are used in combinations for better optimization, i.e., DockThor programs DockTScore [94,98] and blends empirical and force-field-based scoring methods, SMoG2016 [99] fuses empirical and knowledge-based scoring methods, and Galaxy-Dock BP2 Score [100] uses all three: force-field-based, knowledge-based, and empirical scoring methods [94].
- The recent integration of physics-based terms and ML in DockTScore has further enhanced binding energy prediction and conformation ranking [101].
- GNINA docking software, based on an ensemble of convolutional neural networks as a scoring function for scoring the sample view, has outperformed AutoDock Vina [102], once again proving that the paradigm shift from conventional methods to AI-based methods has significantly increased the impartial interpretations of scientific evidence leading to the discovery of targets.

*5.4. Limitations*

Despite its many advantages, the application of AI faces data challenges, such as the data's scale, growth, diversity, and uncertainty. The data sets available for drug development in pharmaceutical companies can involve millions of compounds, and traditional ML tools might be unable to deal with these data types. The recent natural language-based AI tools, such as the GPT4, are anticipated to resolve some of these issues.

While the QSAR-based computational model can quickly predict large numbers of compounds or simple physicochemical parameters, such as log P or log D, predicting biological properties remains challenging. These limitations will be reduced when larger training sets, experimental validation, and more data error training are added in the future.

AI-based data analysis significantly reduces the burden of research and testing in the early discovery phase, as it can handle a large volume of data for profiling molecules. However, it does stand to replace efficacy testing in patients [103] due to the safety issues that form the basis of the regulatory requirements. However, it does significantly decrease the work leading to clinical efficacy testing.

## 6. Structure Modifications

Optimization of the safety and efficacy of drug candidates is a critical step in the drug development process. One approach to optimizing a protein-based drug is to truncate it to enhance its selectivity, potency, and pharmacokinetic–pharmacodynamic properties. Truncation of proteins has been widely used to develop more effective therapeutics and has proven to be a successful strategy in improving bioavailability. Additionally, the optimization of drugs can be combined with reinventing drug strategies to identify new therapeutic uses for existing drugs.

Recently, anti-rheumatoid arthritis effects of native Staphylococcal protein (SpA), recombinant full-length SpA, and a truncated form of SpA were used in a comparative study along with Enbrel (commercial drug) to test reduction in several inflammatory cytokines (IL-8, IL-1$\beta$, TNF-$\alpha$, and IL-6). The truncated SpA had a higher efficacy even when compared to Enbrel. Another study suggested that exogenous truncated inhibitor K562 protein (tIK) has the potential to act as a new therapeutic in patients with Enbrel resistance since their modes of action are contrary to each other [104]. Furthermore, in vivo and silico analyses suggest that the truncated protein resulted in the exposure of the IgG-binding domain, which led to effective binding through an increased radius of gyration [105]. Similar studies have been conducted previously as well.

The N-terminal truncated recombinant form of fibroblast growth factor 21 (FGF 21: amino acids 30-210) demonstrated improved stability and pharmacokinetics in obsess-mouse models. In more than one species of mouse, recombinant FGF21 (Fc-FGF21(RG)) administered once per week produced a similar or higher response than human FGF21 (hrFGF21) administered daily [106,107]. Furthermore, interleukin-2 (IL-2), a cytokine that stimulates the activation of immune cells, has been optimized by truncating the N-terminal region containing a binding site for IL-2 receptor alpha to produce NKTR-214. NKTR-214 has enhanced selectivity and potency, increasing efficacy against tumor cells [108].

## 7. Drug Conjugates

Chemotherapy damages healthy cells that can be protected by using antibody-drug conjugates (ADCs) that direct chemotherapy towards only cancer cells, making it safer [109–111]. The ADCs deliver chemotherapy when a linker connected to a monoclonal antibody binds to a particular target expressed in cancer cells. After binding to the target (cancer protein or receptor), the ADC releases a cytotoxic drug into the cancer cell. "Fully human" monoclonal antibodies engineered to carry human antibody genes are an ideal delivery platform for ADCs. They are highly targeted and cell specific, have an extended circulating half-life, and offer minimal immunogenicity. The chemical "linkers" that combine the antibodies and cytotoxic drugs are highly stable to prevent cleaving (splitting) before the ADC enters the tumor. Anticancer drugs (or "payloads") penetrate the tumor and cause cell death by damaging the DNA of cancer cells or preventing new cancer cells from forming and spreading (Figure 4).

The FDA has approved 14 ADCs, while EMA has approved 8; about 300 are under development [112], mostly for oncological and hematological indications. However, these applications can be expanded to other important disease areas [113]. For example, the payloads for oncology ADCs (oADC) can be derived from natural sources, including the microtubulin inhibitors monomethyl auristatin A MMAE [114], monomethyl auristatin F MMAF [115], mertansine, DNA binder calicheamicin [116], topoisomerase 1 inhibitor SN-38 [117], and exatecan [118].

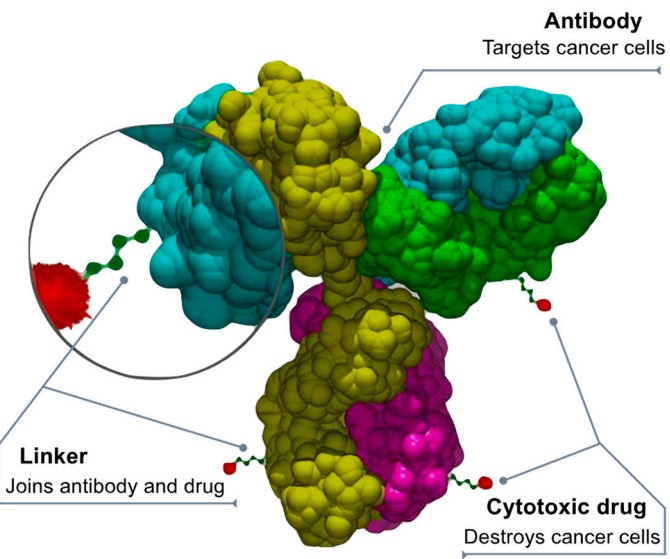

**Figure 4.** Drug-antibody conjugate design. By Bioconjugator—Own work, CC BY-SA 4.0, https://commons.wikimedia.org/w/index.php?curid=58772304 (accessed on 10 April 2023).

Chemical motif-defined linkers include disulfides, hydrazones, peptides (cleavable), or thioethers (non-cleavable). Cleavable and non-cleavable linkers have proven safe in preclinical and clinical trials. The anti-microtubule agent monomethyl auristatin E, or MMAE, a synthetic antineoplastic agent, is delivered to human-specific CD30-positive malignant cells by the enzyme-sensitive cleavable linker in the drug compound brentuximab vedotin. By preventing the polymerization of tubulin, MMAE prevents cell division. MMAE cannot be utilized as a single-agent chemotherapeutic medication due to its severe toxicity. However, the stability of MMAE attached to an anti-CD30 monoclonal antibody is unaffected by extracellular fluid. Trastuzumab emtansine combines the microtubule-formation inhibitor mertansine (DM-1) and antibody trastuzumab, which uses a non-cleavable stable linker [119].

Due to the availability of newer and more robust linkers, the function of the chemical bond has changed. The linker's cleavable or non-cleavable nature determines the cytotoxic medication's characteristics. A non-cleavable linker, for instance, retains the medicine inside the cell. As a result, the entire antibody complex—including the linker and the cytotoxic (anti-cancer) agent—enters the cancer cell that is being targeted, where the antibody is broken down into an amino acid. The resulting complex, which consists of an amino acid, a linker, and a cytotoxic agent, is regarded as an active medication. On the other hand, cleavable linkers are dissociated by cancer cell enzymes. The cytotoxic payload can then leave the targeted cell and destroy nearby cells through a process known as "bystander killing" [120].

AOCs, or antibody-oligonucleotide conjugates, comprise two essential classes of macromolecules: monoclonal antibodies and oligonucleotides. With AOC, various applications, such as imaging, detection, and targeted therapeutics, have profited from the union of the diverse functional modes of oligonucleotides with the potent targeting properties of monoclonal antibodies. The fundamental obstacles to effective ON therapies are cell internalization and absorption. ADCs can be used to get around problems with administering and internalizing ON therapies. The bioconjugation process has been used to obtain several such conjugates.

The utility of ADCs and AOCs is limited to solid tumors because of the larger physical size (150 kDa) since the antibody size cannot be modified [121]. Therefore, nanobody-ON conjugates are intensively used to exploit the small nanobody size to reduce imaging displacement [122].

## 8. Radioimmunoconjugates (RIC)

Radiation is an effective therapy for many tumor types. However, external beam radiation therapy is associated with many nonspecific side effects. Modern radiation techniques such as intensity-modulated and proton beam therapy have increased precision, delivered higher radiation dosages, and reduced toxicities to the surrounding tissues [123] (Figure 5).

Radioimmunotherapy (RIT) has been explored as cancer therapeutics for many decades [124]. RIT utilizes antibodies directed at an antigen expressed on the tumor cell surface to deliver cytotoxic radionuclides that emit α or β particles to the tumor sites. After the radioimmunoconjugates (RICs) bind to the surface antigen on the tumor cells, the α or β particles emitted by the radionuclides induce DNA damage and trigger tumor cell apoptosis [125]. RICs have been viewed mainly as a radiation delivery system to treat metastatic cancer unsuitable for an external beam approach. RICs aim to increase the radiation specificity and allow for the delivery of higher radiation dosages with fewer toxicities. However, the current understanding of tumor immunology suggests that RICs may be more than just a radiation delivery system and present a fertile field for reinventing therapeutic proteins.

Because of their high cytotoxic potential, RICs emitting α- or β-particles can be used for targeted cancer therapy. Cancer treatment using RICs requires careful consideration of the choice of radionuclides and their dosage. β-emitters have a deeper penetration range and a lower linear energy transfer than α-emitters, whereas α-particles can release high energy at a relatively shorter distance. However, while α-particles are more efficient in tumor cell eradication without causing much collateral damage, β-particles are currently most commonly used in radioimmunotherapy. Many β-emitters, such as $^{131}$I and $^{90}$Y, are commercially available and have established techniques for conjugating them to antibodies. For example, $^{90}$Y-ibritumomab tiuxetan and $^{131}$I-tositumomab are US FDA-approved RICs targeting CD20 for treating B-cell non-Hodgkin lymphoma [126]. α-Emitters, on the other hand, are not widely commercially available, techniques for conjugating them to antibodies are not well-established, and pharmacokinetics and dosimetry of α-emitters need further investigation for clinical applications. Large-scale production of radionuclides, especially α-emitters, for clinical applications, requires a significant investment.

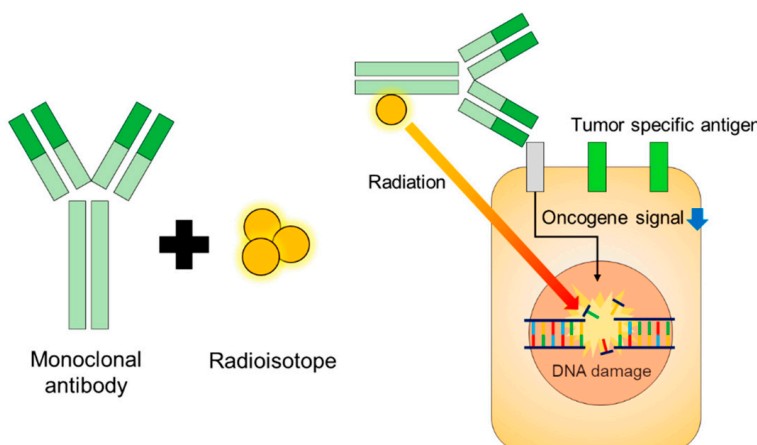

**Figure 5.** Design and function of radioimmune therapies [127].

RICs consist of a targeting antibody conjugated to a radionuclide chelator and indirectly labeled with a radionuclide. The two most commonly used chelators are trans-(*S*, *S*)-cyclohexane-diethylenetriamine pentaacetate (CHX-A''-DTPA) and dodecane tetraacetate (DOTA) [127]. In addition, various radionuclides have been used, including $^{131}$I, $^{111}$In, $^{90}$Y, $^{225}$Ac, and $^{177}$Lu. RICs combine radiation's cytotoxicity with antibodies' specificity to provide powerful antitumor effects to patients with metastatic cancer.

Conventional antibodies directed at intact proteins enable targeting antigens expressed on the surface of tumor cells (Figure 6). If TCR-like antibodies directed at antigen peptides/MHC complexes are used instead, they are also suitable for targeting intracellular antigens. As long as a tumor type is radiosensitive, a wide range of radioisotopes may be chelated to the antibodies, including those emitting α or β emitting particles. An ideal radioisotope would have a short half-life, appropriate penetration range, and high linear energy transfer (LET). In addition to their cytotoxic potential, RICs may be a comprehensive immunotherapeutic agent not limited by the obstacles currently hindering the success of modern cancer immunotherapy. Unlike antibody–drug conjugates, RICs do not require cellular internalization to induce tumor cell kill because of their relatively larger decay sphere of penetration. They circumvent the obstacles related to antigen internalization and uptake of the drug due to lysosomal dysfunction and drug efflux pumps. In this section, we will discuss the wide range of effects of RICs and how they may be harnessed for effective and more specific cancer therapy.

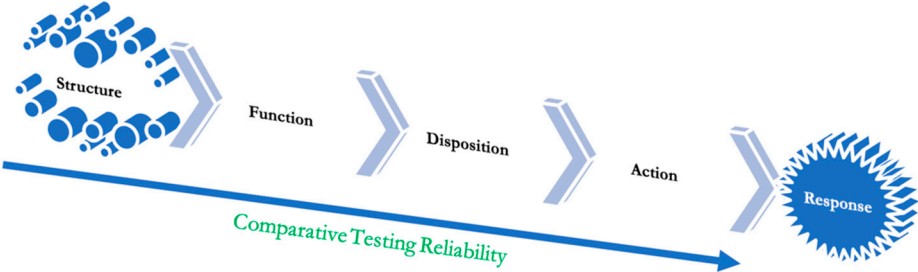

**Figure 6.** The ladder of objectivity from the highest to the lowest.

Only a few active and recruiting studies for non-hematologic solid tumors are registered with Clinicaltrials.gov. The FDA-approved products include Ibritumomab tiuxetan (Zevalin), a monoclonal antibody anti-CD20 conjugated to a molecule that chelates Yttrium-90; Iodine ($^{131}$I) tositumomab (Bexxar) that links a molecule containing Iodine-131 to an anti-CD20 monoclonal antibody, and now withdrawn; and Lutetium ($^{177}$Lu) lilotomab satetraxetan (Betalutin), a combination of lutetium-177 and an anti-CD37 monoclonal antibody.

## 9. Regulatory Perspective

The success of reinventing therapeutic proteins depends significantly on how the regulatory agencies evaluate these products. Sometimes, these are a new class of drugs for which the agencies may need a guideline. In other cases, the agencies may be highly conservative, a mindset that is the responsibility of the developers to change by offering detailed educational discussion in the filing.

The critical reasons for the failure of new drug discoveries include inadequate efficacy or safety, lack of target validation, or inability to meet regulatory requirements. Although computational Drug Design has significantly reduced the chances of riskier drugs entering clinical trials and conserves resources, this should be emphasized in the regulatory filing with justification.

The FDA is leading the perspective of introducing new techniques in structure prediction, target identification, and interaction profiling to revolutionize drug development, setting the industry's standard for precision and efficiency [128]. Recently, these efforts have identified the source of acute kidney injury or hepatic injury from using remdesivir in COVID-19 treatment using a target-prediction software followed by Quantitative-Structure-Activity-Relationship (QSAR) and structure similarity analysis to identify an association between the structure of metabolites and renal-hepatic toxicity [129].

Using AI, the FDA has developed models to classify and clinically monitor organ systems more prone to toxicity [130] and is currently developing natural language processing algorithms to identify molecular targets associated with pediatric cancer through

peer-reviewed literature. In addition, the FDA is conducting research within its Division of Applied Regulatory Science (DARS) program [131].

The DARS is also researching the efficacy of non-clinical methods for anticipating immunogenicity risk. This entails analyzing in vitro assays and cell types, developing in vivo models, and selecting proper controls.

The DARS has also experimented with cutting-edge non-clinical models to forecast cytokine release syndrome, a potentially fatal side effect linked to biological products [132,133], and showed that non-clinical models could effectively demonstrate this adverse event. Furthermore, checkpoint inhibitor oncology therapies for which adverse events cannot be predicted using computational, in vitro, or conventional non-clinical methods can be studied further after successfully demonstrating immune-mediated activation in a non-clinical model [134].

DARS places much emphasis on using molecular target information to anticipate safety issues. Knowing a drug's molecular targets enables early detection of its effects and potential safety issues for new molecules. Still, the exact modeling can also be applied in a comparator mode to study biosimilar candidates. For instance, DARS created several computational techniques, such as machine learning, to forecast a drug's negative effects based on the biological receptors that the drug, or other medications with a similar structure, are known to target [135,136]. These computational methodologies are proving promising in predicting adverse events.

A database for secondary pharmacology activity provided by the industry as part of their application for an investigational new drug is also being built and analyzed by DARS. A drug developer typically performs in vitro target binding and functional assays for 80–100 biological receptors to ascertain potential on-target and off-target effects. However, the targets chosen for the assays and submission format are not currently standardized across the industry. Therefore, data from these assays have been manually extracted and curated into a database to allow easier access and analysis of these study results. Additionally, DARS is engaging in a public–private partnership with the Pistoia Alliance [137] to choose the most effective procedures for submitting these studies to regulatory bodies in the future.

Other issues that the DARS is resolving include using a state-of-the-art alternative to experimental testing to qualify a drug impurity for mutagenic potential [138]. This can significantly help when a biosimilar candidate shows an unmatched impurity. The FDA has suggested using flow imaging microscopy (FIM) to record and analyze images using convolutional neural networks (CNNs or ConvNets) [139]. In addition, the FDA has suggested ParticleSentry AI software [140] to analyze the data to enable protein aggregation profiling.

## 10. Regulatory Submission

Theoretically, the regulatory agencies will treat a reinvented product as a new drug application, and the developer must submit all information required for a new molecule. However, regulatory agencies also allow the submission of information in the public domain, such as the registration dossiers of the selected therapeutic protein from the FDA [141] or EMA [142] portals or the EPARs in the EMA. This leads to a creative approach, "351(a) modified" [143], a term crafted by the authors to significantly reduce the cost and time to approval. Furthermore, even when a therapeutic protein is combined with another drug or a radioactive source, the studies specific to the safety of the therapeutic protein are significantly reduced, making reinventing therapeutic proteins the most efficient and creative path to bringing in new affordable treatment modalities.

### *10.1. Nonclinical Testing*

Figure 6 shows a dependency model leading from receptor binding to patient efficacy. As we move further down the slope, the testing becomes more subjective and less objective, making it a sound argument why a test with higher sensitivity should be reconfirmed with a lesser sensitivity test. Receptor binding remains the most robust and convincing test to

demonstrate the safety and efficacy of therapeutic proteins. The receptor binding need not demonstrate a known pharmacodynamic marker, and the marker must correlate with the clinical response. This relationship forms the basis of the thesis that receptor binding alone can be used to substitute clinical efficacy testing; there is no need for the developers to investigate and find a pharmacodynamic marker either.

The drug approval dossiers and published literature disclose study designs employed in establishing safety and efficacy data of new products; these study models should be replicated for the reinvented product to avoid regulatory approval delays. Further modeling and simulation can provide the dose–response relationships, sensitive dose ranges, population sensitivity, and variability in PD biomarker responses [144–148].

*10.2. Pharmacokinetics–Pharmacodynamics*

Another consideration that can significantly improve the PK/PD data is the inclusion criteria of the test subjects; choosing a narrow characteristic population regarding age, gender, BMI, ethnicity, and pharmacogenomics to antibody responses can significantly reduce the study size and add substantial validity to the data [149].

The PK studies can further support the PD marker utility by extending the data analysis to demonstrate how fast and how much of the parenterally administered drug is leaving the central compartment, thus reaching out to receptor sites; this analysis will demonstrate a similarity in the onset of action. In addition, this property can be compared by adding a pharmacokinetic parameter, the rate of change of distribution volume as a function of time [150], applied in several clinical efficacy comparisons based on clearance and tissue binding [151].

Binding affinities to target antigens can significantly influence the PK of mAbs, requiring measurements of affinity or equilibrium dissociation constant ($K_d$), association rate constant ($k_{on}$), and dissociation rate constant ($k_{off}$). There is an optimal binding affinity beyond which the distribution of the mAb to target tissue may be impaired [152,153]. This affinity is readily established by the characterization of binding to FcRn; as this is a pH-dependent interaction, binding affinity should be measured at pH 6.0 (where FcRn binds mAb in the acidic pH of the endosome) and pH 7.4 (physiological pH where FcRn releases mAb at the cell surface). High binding to FcRn at pH 6.0 and low binding at pH 7.4 is essential for low clearance of mAbs [154,155]. Several studies have investigated the correlation between FcRn binding affinity and the half-life of mAbs, and the contribution of FcRn to prolonging the half-lives of mAbs is well recognized [156]. Since the PK of mAbs depends on PD [157,158], the PK profile projects the PD properties, making it reflective of the PD.

Specifically, pharmacokinetic models should represent physiological variables, and levels of unbound drugs in body fluids should receive greater emphasis [159]. Furthermore, the degree of plasma protein binding, in turn, influences the distribution, action, metabolism, and renal excretion, and most importantly, the distribution triggers that response [160].

$^{14}$C-labeled reworked product testing is an excellent tool to demonstrate changes in the disposition profile, and the FDA highly recommends such studies [161].

For reducing side effects, dose changes can be helpful. These changes are best justified based on the characterization of ADMET (Absorption, Distribution, Metabolism, Excretion, and toxicity). An aphorism written by Nicholas Holford and Lewis Sheiner in 1982, "Pharmacokinetics is what the body does to the drug; pharmacodynamics is what the drug does to the body" [162], fully describes these terms. Pharmacokinetics is the movement of the drug across the membranes of cells, and pharmacodynamics is its interaction with potential biological targets. Collectively, they provide insight into desired therapeutic effects and, sometimes, undesired effects, i.e., toxicity and immunogenic responses. The administered substance goes through a cascade of events inside the body to be efficacious.

Molecular interactions data and the pharmacokinetic–pharmacodynamic (PK/PD) profiles can be used along with AI models to automate the pharmacovigilance process,

pre-clinical and post-clinical surveillance, design efficient clinical trials, suggest the optimal route of administration, and facilitate the selection of highly effective dose regimens.

Discovering and identifying specific binding site poses and affinities results in lower off-target binding, toxicity, and immunogenicity. Preclinical PK/PD analysis, mapping dose–response relationships of exposure, and biological effects in the plasma and target tissue can significantly enhance drug discovery. The effective concentration of the drug in the plasma and the maximum effect is plotted against time, using single or multi-compartment models to characterize PK/PD effects.

PK modeling has proven to be significant in predicting plasma exposure of therapeutics, i.e., if a single 10 mg/kg dose response is known in a mouse model, modeling could help predict the effects of twice-a-day 30 mg/kg dose to hypothesize and optimize a dosing regimen. The PK/PD properties of therapeutic mAbs differ from that of small molecules; hence the concentration of free ligands can be an established marker of their efficacy. Clinically tested effects of galcanezumab dose (120 mg and 240 mg), validated through PK modeling, indicated a steady decrease in the concentration of free ligands resulting in the development of efficacious dose regimens [163]. A PK and target engagement (molecular interaction) study of anti-interferon-γ-induced protein 10 (IP-10) mAb was characterized, which concluded optimal dose strategy and scheduling of drug administration, i.e., approximately eight subcutaneously delivered dose intervals were required weekly in this case to reach steady state [164].

### 10.3. Function Testing

Specialized cell-based bioassays or potency assays, including ELISA, binding assays, competitive assays, cell signaling, ligand binding, proliferation, and proliferation suppression, are essential in ascertaining the mechanism of action and similarity with the parent molecule. On the other hand, functional tests related to the possible MOA, such as apoptosis, complement-dependent cytotoxicity, antibody-dependent cellular phagocytosis, and antibody-dependent cellular cytotoxicity, among others, are necessary but not essential, especially when it is not relevant. For instance, functional tests (ADCC, ADCP, and CDC) are unnecessary for a product that predominantly targets a soluble antigen [165–169].

Thus, comparable bioassay results should be sufficient when PD markers are unavailable, such as for mAbs. Therefore, a complete bioassay toolbox is a crucial enabler for applying the proposed clinical development paradigm. The toolbox requires multiple assays, ideally cell-based, to cover all relevant functions of a molecule with accurate and precise quantitative readouts and agreement with the regulators on the bioassay designs, including their validation [170,171]. For example, comparable binding affinities to TNF-α, C1q complement, and a complete panel of Fc-receptors for etanercept have proven sufficient to establish biosimilarity since this binding is the primary mechanism of action of etanercept [172].

For a product with multiple biological activities, a set of relevant functional assays designed to evaluate the range of activities of the product can be tested. For example, specific proteins possess multiple functional domains that express enzymatic and receptor-binding activities. Potency is the measure of biological activity. When immunochemical properties are part of the activity attributed to the product (for example, antibodies or antibody-based products), analytical tests to characterize these properties are readily available.

### 10.4. Immunogenic Response

Proteins are immunogenic and capable of producing neutralizing antibodies (NAbs) that bind to drug products and may diminish or eliminate the associated biological activity; these are unintended and undesirable outcomes. Standard immunoassays can detect drug-specific antibodies but cannot distinguish NAbs. Therefore, cell-based assays are often preferred because they closely mimic the mechanism by which NAbs and drug products interact in vivo. However, each cell-based NAb assay is unique and based on several factors, such as the drug product, study population, and development phase (preclinical or clinical).

In addition, the type of NAb assay (direct or indirect) depends on the drug's mechanism of action. Generally, the appearance of NAbs is not a pivotal issue if their presence does not alter the disposition profile, such as in the case of insulin [173]. Reinvented products should be compared with the original product to ensure that the changes made, either in structure or combination compositions, do not alter the NAb level or immunogenicity.

## 11. Conclusions

The higher attrition rate of new drug discovery from conventional methods leads to a wastage of resources and time after hefty preclinical and clinical testing [174]. As a result, the cost of new drug development has skyrocketed over the past decade into billions of dollars [175]. Compared to chemical drugs, therapeutic proteins present a remarkable opportunity to reinvent their use because of their mechanism of action—receptor binding—and vast structure that presents hundreds of possibilities for finding new uses of an approved therapeutic protein. Billions of dollars of markets are thus available without spending the billions and providing new therapies at a much lower cost when the approved therapeutic proteins are put into a reinvention cycle. This exercise was much more difficult until a decade ago when AI and ML systems entered the field of science. High-throughput screening enabled identifying potential targets using in silico approaches. As a result, the regulatory burden of the reinvented products is substantially less than a new molecule, and so are the risks of failure.

It is strongly urged that developers, both large and small, investigate this remarkable treasure of therapies available to explore at a highly affordable cost and bring therapies for thousands of rare and complex diseases.

**Author Contributions:** Both authors contributed equally to conceptualization, data curation, writing, review, and editing. All authors have read and agreed to the published version of the manuscript.

**Funding:** This research received no external funding.

**Institutional Review Board Statement:** Not applicable.

**Informed Consent Statement:** Not applicable.

**Data Availability Statement:** Not applicable.

**Conflicts of Interest:** The author declares no conflict of interest.

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
