# Peer review of "Reinventing Therapeutic Proteins: Mining a Treasure of New Therapies"

_biologics, doi:10.3390/biologics3020005_

Round 1
Reviewer 1 Report
The present article describes the possibilities of machine learning and artificial intelligence as auxiliary tools for searching for potential drugs. The article is well structured, with visual illustrations of the text and relevant references.
As a recommendation to the authors, could you please indicate what are the main limitations of the use of AI and ML in drug development? Which steps of the "traditional" preclinical and clinical parts could be replaced and which ones will remain unchanged?
Some abbreviations are not deciphered at the first mention. This article deserves to be printed.
Author Response
Thank you very much for your constructive comments that have helped improve our writing.
As a recommendation to the authors, could you please indicate what are the main limitations of the use of AI and ML in drug development? Which steps of the "traditional" preclinical and clinical parts could be replaced and which ones will remain unchanged?
Several paragraphs are added to point out the limitations as well as a perspective on how these may be removed in the past.
Some abbreviations are not deciphered at the first mention. This article deserves to be printed.
We have added the description when an abbreviation is used for the first time.
Reviewer 2 Report
It is a comprehensive and well written review manuscript covering from the basics of therapeutic proteins through the strategies for regulatory submission. However, two points are proposed for minor revision:
(1) Relatively little is described on technological approaches of how AI/ML and HTS are applied for drug reinventing. This needs to be reinforced.
(2) It is suggested that rather general descriptions on protein folding, antibodies, ADCs, and conjugates, that are not particularly relevant to drug reinventing, need to be condensd or omitted.
[END]
Author Response
- Relatively little is described on technological approaches of how AI/ML and HTS are applied for drug reinventing. This needs to be reinforced.
Several paragraphs are added to show the modality of use of AI/ML and HTS
Thanks for your editing comments.
It is suggested that rather general descriptions on protein folding, antibodies, ADCs, and conjugates, that are not particularly relevant to drug reinventing, need to be condensed or omitted.
We have removed some parts but felt the need to leave a few basic elements to enable engaging novices in the field to benefit. Hope you find it acceptable.